# Zika virus infection in pregnancy: a protocol for the joint analysis of the prospective cohort studies of the ZIKAlliance, ZikaPLAN and ZIKAction consortia

A E Ades,[1] Elizabeth B Brickley,[2] Neal Alexander,[2] David Brown,[3] Thomas Jaenisch,[4] Demócrito de Barros Miranda-Filho,[5] Moritz Pohl,[6] Kerstin D Rosenberger,[4] Antoni Soriano-Arandes [ID],[7] Claire Thorne [ID],[8] Ricardo Arraes de Alencar Ximenes,[5] Thalia Velho Barreto de Araújo,[5] Vivian I Avelino-Silva,[9] Sarah Esperanza Bethencourt Castillo,[10] Victor Hugo Borja Aburto,[11] Patrícia Brasil,[12] Celia D C Christie,[13] Wayner Vieira de Souza,[14] Jose Eduardo Gotuzzo H,[15] Bruno Hoen,[16,17] Marion Koopmans,[18] Celina Maria Turchi Martelli,[14] Mauro Martins Teixeira,[19] Ernesto T A Marques,[14,20] Maria Consuelo Miranda,[21] Ulisses Ramos Montarroyos,[22] Maria Elisabeth Moreira,[23] J Glenn Morris,[24] Barry Rockx,[18] Paola Mariela Saba Villarroel,[25] Carmen Soria Segarra,[26,27] Adriana Tami [ID],[10,28] Marília Dalva Turchi,[29] Carlo Giaquinto,[30] Xavier de Lamballerie,[31] Annelies Wilder-Smith,[32] EC Zika Consortia Vertical Transmission Study Group

**Correspondence to**
Prof Claire Thorne;
claire.thorne@ucl.ac.uk

## ABSTRACT

**Introduction** Zika virus (ZIKV) infection in pregnancy has been associated with microcephaly and severe neurological damage to the fetus. Our aim is to document the risks of adverse pregnancy and birth outcomes and the prevalence of laboratory markers of congenital infection in deliveries to women experiencing ZIKV infection during pregnancy, using data from European Commission-funded prospective cohort studies in 20 centres in 11 countries across Latin America and the Caribbean.

**Methods and analysis** We will carry out a centre-by-centre analysis of the risks of adverse pregnancy and birth outcomes, comparing women with confirmed and suspected ZIKV infection in pregnancy to those with no evidence of infection in pregnancy. We will document the proportion of deliveries in which laboratory markers of congenital infection were present. Finally, we will investigate the associations of trimester of maternal infection in pregnancy, presence or absence of maternal symptoms of acute ZIKV infection and previous flavivirus infections with adverse outcomes and with markers of congenital infection. Centre-specific estimates will be pooled using a two-stage approach.

**Ethics and dissemination** Ethical approval was obtained at each centre. Findings will be presented at international conferences and published in peer-reviewed open access journals and discussed with local public health officials and representatives of the national Ministries of Health, Pan American Health Organization and WHO involved with ZIKV prevention and control activities.

### Strengths and limitations of this study

► This will be a pooled analysis of data from three international consortia conducting prospective cohort studies of outcomes following Zika virus (ZIKV) infection in pregnancy in 20 centres in 11 countries.

► Standardised definitions of outcomes will provide clarity about the absolute risks of adverse outcomes, which have not been reported consistently in prospective studies so far.

► These studies include a control group of women with no evidence of ZIKV infection in pregnancy, allowing improved estimation of the proportion of adverse events attributable to ZIKV in pregnancy.

► Inferences will be limited by: difficulties in distinguishing between women who did and who did not experience a ZIKV infection in pregnancy, due to the high frequency of mild and asymptomatic infections and the low sensitivity and specificity of diagnostic tools; low diagnostic sensitivity of markers of congenital infection; and outcome data that are not missing at random.

## INTRODUCTION

Following the emergence of Zika virus (ZIKV) in Asia and the Pacific in 2013[1] and the subsequent introduction to Brazil,[2] clusters of neonates with severe neurological

complications and microcephaly were observed across Latin America. Following recent experiences with the H1N1 influenza pandemic and Ebola outbreak in Western Africa, the need for coordinated international research on ZIKV was quickly recognised. In January 2016, before WHO declared a Public Health Emergency of International Concern,[3] the European Commission (EC) issued a funding call to set up a network in Latin America and the Caribbean with the aim of implementing and coordinating urgently required research, while simultaneously contributing to research capacity and preparedness for other emerging infectious diseases. Three consortia were funded: ZIKAlliance (https://zikalliance.tghn.org/),[4 5] ZikaPLAN[6] (https://zikaplan.tghn.org/) and ZIKAction (http://zikaction.org/). All are multidisciplinary international collaborations with active investigations in epidemiology, virology, immunology, diagnostics, mathematical modelling, social science and animal studies. Each consortium includes its own prospective cohort study of ZIKV in pregnancy and a shared work package that aims to ensure the harmonisation of protocols and data sets in order to facilitate a pooled analysis of cohort data. The primary aim of the pooled analysis is to investigate the incidence of adverse outcomes of ZIKV infection in pregnancy, including 'congenital infection, microcephaly, Zika congenital syndrome and other sequelae of ZIKV infection'.

The aim of this paper is to present a protocol for this pooled analysis. Data have been or are still being collected in multiple sites in 20 regional coordinating centres spread over 11 countries and regions across Latin America and the Caribbean. There are 15 ZIKAlliance centres: Sao Paulo, Rio de Janeiro, Recife and Belo Horizonte (Brazil); Valencia (Venezuela); Bucaramanga (Colombia); Guayaquil (Ecuador); Lima (Peru); Jalisco, Nayarit, Veracruz, Yucatan (Mexico); Santa Cruz de la Sierra (Bolivia); Havana (Cuba); Guadeloupe (French Territory of the Americas); 3 ZikaPLAN centres: Goiânia, Rio de Janeiro, Recife (Brazil) and 2 ZIKAction centres: Kingston (Jamaica) and Port-au-Prince (Haiti). Recruitment to ZIKAlliance began May 2017, December 2015 for ZikaPLAN, and September 2017 for ZIKAction. Over 700 women with confirmed infection had been recruited by April 2020.

Several studies of ZIKV in pregnancy have recently been published. In registry-based studies,[7–9] fetuses and newborns of women with confirmed infection in pregnancy have been reported to have 'potentially Zika-related' adverse outcomes at rates of up to 15%, with higher risk of Zika-associated adverse outcomes in the first trimester. Registry-based studies are likely to overestimate the risk of severe clinical manifestations and underestimate the risk of more mild clinical presentations because they recruit both prospectively ascertained ZIKV-infected pregnant women and women whose infection was recognised retrospectively following the birth of an infant with congenital abnormalities. Prospective studies of congenital infection have variously reported 25% 'severe' and 21% 'mild to moderate' outcomes in French Guiana,[10] and 27% adverse outcomes in Brazil.[11] The specificity of these outcome definitions for ZIKV in pregnancy is not known as these studies did not include a control group of women with no ZIKV infection in pregnancy. In another Brazilian study, the risk of adverse outcomes was reported to be 46% in births to women with NAAT (Nucleic Acid Amplification Test)-confirmed ZIKV infection in pregnancy compared with 11.5% in NAAT-negative women.[12] In a large prospective study based in the French Territories of the Americas, among infants born to women with NAAT-confirmed ZIKV infection, 7.0% presented with neurologic or ocular birth defects and 3.1% met the study's criteria for Congenital Zika Syndrome (CZS),[13] which is characterised by several unique features.[14] An important limitation in the comparison of the results of these different studies is the lack of a standard definition of CZS and of the clinical and diagnostic procedures used to evaluate these children, leading to possible misclassification of the outcomes studied.

The vertical transmission rate is the probability of congenital infection in births to women with infection in pregnancy. The rates reported so far, 26%[11] and 35%,[10] are based on laboratory markers of congenital infection such as NAAT or IgM in the fetus or newborn. However, a prospective cohort retrospectively reconstructed from a register study estimated the vertical transmission rate to be only 9%.[15] Comparison of these rates is difficult as different markers and different biological samples were used. In addition, although these tests (NAAT and IgM) are analytically sensitive and specific they have poor diagnostic sensitivity as markers of congenital infection. These markers were absent from serum in a high proportion of CZS cases[16 16 17] and in newborns with other potentially ZIKV-related adverse outcomes born to women with confirmed ZIKV during pregnancy.[10 11] Clearance of virus from amniotic fluid and fetal blood has been reported in cases of CZS, even when ZIKV is found in brain tissue postmortem.[18 19] It, therefore, appears that fetal infection may occur, causing profound damage, but clearing before delivery and leaving no discernable immunological trace in serum. Consequently, in this study, we will document the prevalence of markers of congenital infection using uniform criteria, recognising that this is an underestimate of the true vertical transmission rate.

Regarding effect modifiers, a number of studies have reported a higher incidence of congenital abnormalities following maternal infections in the first trimester.[7 10 11 14] Maternal symptoms during acute ZIKV infection do not appear to be a risk factor for adverse outcomes.[20] There is evidence of antibody dependent enhancement of ZIKV by dengue virus (DENV) antibody in animal models,[21] but it is unclear whether previous DENV infection or exposure to other flaviviruses has a protective, risk-enhancing or null effect, in maternal or congenital infection in humans.[22] It also remains to be established whether a previous ZIKV infection confers protective immunity.

Little is currently known about risk factors for transplacental transmission of ZIKV.

The analysis plan described here complements the recently published protocol of the ZIKV Individual Participant Data (IPD) Consortium,[23] which will eventually include data from the three EC consortia as well as data from many other sources. Although the objectives of the protocols are similar, different methods are proposed in relation to design of included studies, definition of congenital infection, and approach to imperfect diagnosis of maternal infection.

In light of the unexplained heterogeneity in reported rates of adverse outcomes, and the variation in prevalence of markers of congenital infection, a pooled analysis of data from 20 centres following similar protocols with harmonised definitions of clinical and laboratory outcomes will provide important new information on outcomes of ZIKV in pregnancy.

## OBJECTIVES OF THE JOINT ANALYSIS

1. To estimate the risk of adverse outcomes in the fetus, newborn, and child following maternal ZIKV infection in pregnancy, compared with outcomes in controls with no evidence of maternal infection in pregnancy (MIP).
2. To estimate the prevalence of markers of congenital infection among fetuses and liveborn infants following maternal ZIKV infection during pregnancy.
3. To assess the associations between trimester of maternal infection, presence or absence of maternal symptoms, and previous flavivirus infections with adverse outcomes and markers of congenital infection.

## METHODS
### Participants
Pregnant women were eligible only if their infection status during pregnancy (infected or not infected) was ascertained prior to the detection of adverse outcomes, or was not influenced by fetal examination or outcome on delivery. This definition is compatible with retrospective testing of previously collected maternal samples, after delivery. Although the unit of recruitment is the mother, the unit of analysis is the fetus, newborn and infant; multiple births are sufficiently rare to be treated as independent observations.[24]

### Study design
Consenting women were screened in pregnancy for markers of ZIKV infection. Those in whom MIP was suspected were followed with enhanced investigations. In ZIKAction and ZIKAlliance, all deliveries to these women, including fetal losses, stillbirths and newborns were examined clinically and tested for markers of congenital infection. This testing was not routinely performed in ZikaPLAN. In all three cohorts, newborns were prospectively followed to identify any adverse outcomes that may

develop later. In all three consortia, a sample of newborns delivered to women with no evidence of infection in pregnancy served as an unexposed control group.

There were some differences between the protocols adopted by the three consortia in terms of how women were recruited into the study, and the choice and scheduling of tests and investigations (online supplemental table S1). In ZIKAction and ZIKAlliance, women were recruited regardless of symptoms during pregnancy, although report of symptoms was recorded. In ZikaPLAN, only women with rash, a common sign of ZIKV infection, were recruited. Statistical analyses will therefore be stratified by whether the mother reported symptoms in pregnancy.

When the studies were designed, there was little information on the risk of adverse outcomes of ZIKV in pregnancy, on vertical transmission rates, nor on what infection rates among pregnant women might be expected. Formal sample size calculations were not undertaken.

### Patient and public involvement statement
There was no patient or public involvement in this study.

### Target parameters and terminology of vertical transmission studies
Six categories of joint congenital infection status and maternal infection status (A–F) are defined in table 1, which illustrates the logic of an idealised prospective study. The usual target parameters are the vertical transmission rate, which is the probability of congenital infection following MIP, (A+B) / (A+B+C+D); and the rate of adverse outcomes in those with congenital infection, A/(A+B). The definitions of 'adverse outcomes', congenital infection and MIP will be determined by a Joint Diagnostics Group and a Joint Endpoint Review Group, after the data have been assembled. Estimates of these parameters are standard in the classic studies of vertical transmission of HIV,[25 26] toxoplasmosis[27–29] and cytomegalovirus.[30] In studies of less specific outcomes, the event rate C/(C+D) in fetuses and newborns of women with MIP but in whom no congenital infection occurred (paediatric control group 1 in table 1) forms a comparison group[31 32] representing the adverse event rate that is due to MIP in the absence of congenital infection. The present analysis plan is modelled closely on these earlier studies, but includes adaptations to take account of the difficulties in diagnosing maternal and congenital ZIKV infection.

For example, because cases of congenital infection cannot be reliably identified by diagnostic tests, we can only estimate the prevalence of laboratory markers of vertical infection (ie, NAAT or IgM) (Objective 2). Similarly, the 'overall' (unconditional) adverse event rate is taken as the primary outcome for objective 1; this includes all births to women with MIP, (A+C)/(A+B+C+D) (table 1). All three consortia included a further control group of births to unexposed women, those with no infection in

**Table 1** Design of an idealised prospective vertical transmission study

| | | | Maternal infection status | |
| --- | --- | --- | --- | --- |
| | | | **MIP** | **No MIP** |
| Congenital Infection Status | congenital infection | Adverse outcomes | A | 0 |
| | | No adverse outcomes | B | 0 |
| | No congenital infection | Adverse outcomes | C | E |
| | | No adverse outcomes | D | F |

The vertical transmission rate is estimated by (A+B)/ (A+B+C+D). The rate of adverse outcomes conditional on congenital infection is A/(A+B). This can be compared with the rate of adverse outcomes in newborns with no congenital infection, C/(C+D), who form a control group (Paediatric control group) to account for potential confounders associated with maternal infection. An overall, non-conditional estimate of the adverse eventrate is (A+C)/(A+B+C+D). Follow-up of births to women with no MIP creates a second control group (maternal control group), in which the rate of adverse outcomes, E/(E+F), can be compared with the rate in births with no congenital infection to womenwith MIP. Estimated effects of MIP based on the maternal control group are vulnerable to confounding by factors associated with MIP. Some cells are set to zero as there can be no congenital infection without MIP.

MIP, maternal infection in pregnancy.

pregnancy (Maternal Control Group in table 1). The adverse event rates in this group, E/(E+F), represents a baseline for comparison with the overall event rates in exposed women.[33]

Although estimates of the vertical transmission rate are compromised, it may still be of interest to compare adverse outcome rates in the MIP with congenital infection group (ie, A/(A+B)), the MIP without congenital infection group (ie, C/(C+D)) and the No MIP group (ie, E/(E+F)), as this may provide insight into whether adverse fetal outcomes are associated with MIP in the absence of demonstrable fetal infection.

In addition to the challenges associated with the laboratory definition of congenital infection, it is also difficult to discriminate between pregnancies with MIP and with No MIP as required in analyses based on table 1. A positive NAAT result or seroconversion during pregnancy are sufficient to confirm MIP. However, even if tested per protocol, MIP may be missed due to the narrow window of detection of NAAT tests, perhaps as low as 14 days.[34] Tests of recent infection, including IgM, IgG3 or avidity assays, may reflect infection during pregnancy, but may also be the result of infection prior to pregnancy,[35] and immunological cross-reactivity to DENV antibody may need to be ruled out.[36] These tests, therefore, indicate suspected but not confirmed MIP. An IgG negative response in the woman or newborn at or tests, perhaps as low as 14 days.[34] An IgG negative response in the woman or newborn at or shortly after delivery can be taken as suggestive of No MIP, although the dynamics of ZIKV IgG are not well documented. While a negative IgG is likely to be uncommon in some sites, as some women will have experienced a ZIKV infection prior to pregnancy, it lowers the probability that a ZIKV infection would have occurred in pregnancy. Taking all this together, MIP status will be characterised as 'Confirmed', "Suspected, 'No Evidence of MIP' (ie, all NAAT and IgM tests negative), and 'No MIP' (ie, all NAAT and IgM tests negative and IgG negative at or shortly after delivery). To maintain the principle of prospective ascertainment, confirmation of congenital infection will have no impact on the mother's assigned infection status.

### Definitions of variables

To support the pooled analyses, a Joint Diagnostics Group consisting of immunologists and virologists and a Joint Endpoint Review Group consisting of paediatricians with experience of congenital ZIKV will be convened to agree on standardised case definitions. The names and qualifications of members of both groups will be published at the time of the joint data analysis, together with the rationale and process for their recruitment. Online supplemental table S2 provides some provisional definitions for: MIP (Confirmed, Suspected, No Evidence of MIP, No MIP), laboratory markers of congenital infection (present or absent), signs and symptoms compatible with CZS, other potentially Zika-related outcomes (OPZRO) and trimester of MIP. The Joint Diagnostics Group will also have responsibility for definitions of covariates, such as previous flavivirus infection. Definitions of both diagnostic categories and clinical endpoints will be based on the best information available at the time of analysis, and will, to the greatest extent possible, be harmonised across consortia and across centres within consortia. As the expert groups will be assembled from representatives of each consortium, they will already have examined the data prior to a joint analysis, so that blinding will not be possible; however, they will devise diagnostic and clinical criteria that can be applied objectively across the three consortia.

An essential aspect of the definitions for MIP status used in the statistical analyses below is that they must be based exclusively on the prospective diagnostic testing. For example, although CZS and/or laboratory evidence of congenital infection in the newborn provides compelling evidence of MIP, this would not affect the MIP status as ascertained prospectively. Thus, we expect to observe some newborns with CZS and/or with laboratory markers of congenital infection delivered to women with Suspected MIP, or even those categorised as having no evidence of MIP.

## STATISTICAL ANALYSIS
### Analysis of data from individual centres
#### Descriptive tables

We will produce descriptive tables providing a breakdown of the numbers in each centre with confirmed MIP, suspected MIP, no Evidence of MIP and no MIP, and the type of evidence on which this is based (eg, NAAT, seroconversion, tests of recent infection) (online supplemental table S3). The similar online supplemental table S4 will document numbers with signs or symptoms compatible with CZS and with markers of congenital infection, and the type of evidence on which this is based, for example NAAT, IgM or clinical markers (CZS).

#### Objective 1: adverse outcomes

Prospectively ascertained MIP status will be tabulated against overall (ie, unconditional on congenital infection status) adverse outcomes (table 2). Various risks can be estimated within each MIP category, including: the risk of signs and symptoms compatible with CZS, the risk of OPZRO, the risk of both combined, of individual signs and symptoms or of signs and symptoms grouped in clinically (eg, ophthalmological defects) or embryologically meaningful ways. Outcomes may be binary (eg, microcephaly) or continuous (eg, head circumference) or multicategory (CZS-related outcomes, OPZRO, asymptomatic). As well as congenital anomalies, rates of standard outcomes, in the absence of congenital anomalies, will be documented, including: fetal loss, stillbirth, low birth weight, intrauterine growth retardation (IUGR) and premature delivery.

The rate of adverse outcomes in the no MIP group represents the study-specific background rate of adverse outcomes (ie, in the absence of MIP, cntrol group 2), to be compared with rates in women with confirmed MIP. Absolute risks will be estimated as well as risk ratios and risk differences.

Parallel sets of estimates will be calculated in the suspected MIP and no evidence of MIP groups, as particular adverse events indicate lack of diagnostic specificity and sensitivity in the testing protocol during pregnancy and are therefore informative regarding the effectiveness of the maternal testing protocol in each centre; for example, microcephaly in the No Evidence of MIP group would indicate a lack of sensitivity. These sets of estimates may be pooled, respectively, with adverse outcome risks in Confirmed MIP and No MIP groups in sensitivity analyses.

#### Objective 2: laboratory and clinical markers of vertical transmission

Estimates of the prevalence of markers of congenital infection will be produced in all centres with available data for each MIP group except the no MIP group, as the definition of No MIP is not compatible with laboratory markers of congenital infection. For this purpose, the numerator will be the number with markers of congenital infection, and the denominator will be the sum of the numbers with and without markers of congenital infection. Separate estimates will be obtained for: laboratory markers, clinical markers (namely CZS), and combined laboratory and clinical markers (table 3). Rates in the no evidence of MIP group are of interest as they carry information about the diagnostic accuracy of the maternal testing protocol.

#### Objective 3: effect of covariates

Separate estimates of the prevalence of adverse pregnancy outcomes, and the probability of laboratory and clinical markers of congenital infection, will be produced for each trimester of maternal infection, and by presence or absence of maternal symptoms.

Analyses of adverse event frequencies and of markers of congenital infection can be extended to include multiple covariates, using logistic regression. These might include potential effect modifiers, such as previous arbovirus infection or coinfection, or confounding factors such as socioeconomic indicators likely to be associated with both arbovirus exposure and adverse outcomes. However, at the time of writing it is not known whether sufficient data will be available for regression analyses.

#### Secondary objectives

ZIKV infection in pregnancy could lead to adverse pregnancy and birth outcomes either following a congenital infection (cell A in table 1) or in the absence of congenital infection (equally in cells A and C). Although absence of markers of congenital infection does not

---

**Table 2** Scheme for a generic analysis of risks of adverse outcomes by prospectively ascertained MIP status

| Centre | MIP | | | |
| --- | --- | --- | --- | --- |
| | **Confirmed** | **Suspected** | **No Evidence of MIP** | **No MIP** |
| Symptom 1 | | | | |
| Symptom 2 | | | | |
| Symptom 3 | | | | |
| : | | | | |
| No symptoms | | | | |
| Total | | | | |

Presence or absence of fetal and neonatal signs or symptoms (eg, microcephaly, brain calcifications, arthrogryposis) and other potentially Zika-related outcomes (eg, fetal loss) in the different MIP groups will be compared.
MIP, maternal infection in pregnancy.

**Table 3** Scheme for generic analysis of markers of congenital infection by prospectively ascertained MIP status

| Centre | MIP | | | |
| --- | --- | --- | --- | --- |
| | Confirmed | Suspected | No Evidence of MIP | No MIP |
| Both laboratory and clinical markers of congenital infection | | | | |
| Only laboratory markers of congenital infection | | | | |
| Only clinical markers of congenital infection | | | | |
| No markers of congenital infection | | | | |
| Not tested | | | | |
| Total | | | | |

Markers of congenital infection may include laboratory markers, clinical markers, such as microcephaly or both combined.
MIP, maternal infection in pregnancy.

rule out congenital infection, we might still expect to observe more adverse outcomes in fetuses and newborns with laboratory markers of congenital infection if those outcomes are caused by congenital infection. By contrast, adverse outcomes that are the result of MIP in the absence of congenital infection (Cell E in table 1) should occur equally with or without laboratory markers of congenital infection. Based on literature on other infections in pregnancy, including DENV,[37] adverse outcomes associated with MIP in the absence of congenital infection potentially include: fetal loss, stillbirth, prematurity, IUGR and low birth weight for gestational age.[38] The analysis would be based on a tabulation of presence or absence of neonatal symptoms, or sets of symptoms (online supplemental table S5), and would be stratified by trimester of maternal infection, as this is likely to be associated with the presence of markers of congenital infection and with adverse outcomes.

### Missing covariates
All analyses will be conducted on a 'complete case' basis, in the first instance. Methods for handling missing covariates, such as imputation,[39] will be considered after the extent and patterns of missing data have been explored.

### Combining data across centres
The above analyses will generate a series of centre-specific estimates of proportions, relative risks comparing MIP and No MIP groups, risk-differences and means of continuous variables, stratified by trimester and maternal clinical presentation. If logistic regression is used to examine effect modifiers and confounders, further estimates of interaction terms or adjusted estimates can also be produced.

All these estimates can be combined across centres using fixed or random effects models in a 'two-stage' meta-analysis. Bayesian Markov chain Monte Carlo methods will be used as this will facilitate the use of exact binomial and multinomial likelihoods, which have a better performance with low and zero cells counts. Vague priors will be employed. Centre-specific random effect estimates will be sampled from beta distributions for binomial outcomes data, Dirichlet distributions for multinomial data and normal distributions for continuous data. We will report ranges, between-centre SD, mean effects and predictive effects with 95% credible intervals for each estimate.

In combining estimates from different centres, we will take account of the fact that in ZikaPLAN only women with rash were recruited, so that women with no ZIKV infection may have experienced other exanthematic infections,[12] including arbovirus infections such as dengue and chikungunya, which may themselves be associated with adverse outcomes.[40 41]

Depending on the results of two-stage analyses, and the completeness of covariate data, an individual patient data one-stage meta-analyses will be considered for each objective, as a secondary or sensitivity analysis, with centre as an additional fixed 'intercept' term.

### Sensitivity analyses
#### Definitions of MIP status, ZIKV-related outcomes and laboratory markers of congenital infection
We will report differences in adverse event rates between Confirmed MIP and Suspected MIP, and between No Evidence of MIP and No MIP. If the differences are small, we will produce results pooling these categories as a sensitivity analysis.

Further, the No Evidence of MIP category can be subdivided into women who were tested per protocol and those who may have been tested less completely. The impact of compliance with protocols will be explored, as it is expected to impact on the proportion of women with MIP who are classified as 'no evidence of MIP' and, hence, on the probability of observing adverse outcomes in this group. Similarly, we will conduct sensitivity analyses around the definitions of Confirmed and Suspected MIP on advice from the Joint Diagnostics Group.

Alternative sets of estimates will be generated using alternative criteria for CZS-related outcomes and OPZRO, that are more, or less, specific for ZIKV in pregnancy. Similarly, we will explore the impact of varying the laboratory criteria for congenital infection on the advice of the Joint Diagnostics Group.

#### Independent ascertainment of outcomes
A critical requirement of all these analyses is that the ascertainment of markers of congenital infection status and clinical outcomes in the fetuses and newborns, and

developing infant are all independent of each other, and also independent of MIP status. For example, ideally the same laboratory testing for congenital infection is carried out regardless of whether the pregnancy outcome is a fetal loss, termination of pregnancy, stillbirth, a case of CZS, or an apparently healthy asymptomatic infant.

These assumptions are difficult, and in certain respects (eg, first trimester fetal loss and terminations) not possible, to fully implement in practice. To address these inevitable limitations in the analysis, which are expected to impact more on objective 2 than on objective 1, we will carry out exploratory analyses aimed at detecting potential deviations from protocol. For example, the distribution of trimester of MIP should not be associated with prospectively ascertained MIP status. Guided by the results, we will carry out sensitivity analyses that make a range of assumptions about the distribution of missing data, especially data on markers of congenital infection. A series of scenarios will be examined to assess robustness of results to inherent and/or unplanned deviations from the ideal protocols required for unbiased estimation of the target parameters.

## DISCUSSION

Risks of CZS and other adverse birth outcomes of ZIKV infection in pregnancy can only be assessed through studies that recruit women whose infection status is prospectively ascertained, or, if retrospective, ascertainment is independent of outcomes. However, reported risks of adverse outcomes even from prospective studies have been highly variable,[10–13] as have vertical transmission rates based on laboratory markers of congenital ZIKV infection.[10 11 15] An important role for joint analyses of multiple studies is to explore whether this heterogeneity in outcomes can be explained by individual or study-level covariates. To do this, it is essential that incidental sources of variation, such as those arising from differences in outcome reporting or diagnostic testing, are controlled or eliminated as much as possible. One of the most difficult sources of variation between consortia, and between sites within consortia, lies in diagnosis of maternal infection. Our approach is to have an Expert Diagnostics Group produce a harmonised classification of confirmed MIP, suspected MIP, no Evidence of MIP and no MIP, and to compute a range of estimates of the relative effect of maternal infection on outcomes, grouping these in different ways. An analysis based on the confirmed MIP and no MIP groups alone would be expected to generate the largest estimates of relative effect, because both poor sensitivity and poor specificity will tend to bias effect estimates towards the null.

An alternative proposal[23] in relation to maternal infection status is to treat test sensitivity and specificity as study-level covariates in a meta-regression, but the risk of false positive diagnosis depends more on the incidence of ZIKV and cross-reacting antibodies to other arbovirus infections such as dengue than on test specificity. The ZIKV IPD Consortium protocol differs in two other ways. First it proposes to include surveillance studies, which may result in overestimating the risk of adverse outcomes due to retrospective ascertainment of infected women following adverse newborn outcomes.[7 9] Second, congenital infection is to be defined by clinical and radiological criteria alone.

Other statistical methods may have been developed by the time the data becomes available for these analyses. Whatever form of analysis is adopted, a standardised pooled analysis from three large consortia comprising 20 centres will provide valuable information about these parameters, which will assist in framing a public health response and advice to women who might be exposed in future. It may also throw light on pathological mechanisms leading to adverse outcomes, which could help in the development of therapeutic or prophylactic interventions.

**Author affiliations**
[1]Department of Population Health Sciences, University of Bristol Medical School, Bristol, UK
[2]Department of Infectious Disease Epidemiology, London School of Hygiene and Tropical Medicine, London, UK
[3]Flavivirus Reference Laboratory, Fundacão Oswaldo Cruz, Rio de Janeiro, Brazil
[4]Department of Infectious Diseases, Section Clinical Tropical Medicine, UniversitatsKlinikum Heidelberg, Heidelberg, Germany
[5]Universidade Federal de Pernambuco, Recife, Brazil
[6]Institute of Medical Biometry and Informatics, University of Heidelberg, Heidelberg, Germany
[7]Paediatric Infectious Diseases and Immunodeficiencies Unit, Hospital Universitari Vall d'Hebron, Vall d'Hebron Research Institute, Barcelona, Spain
[8]Great Ormond Street Institute of Child Health, University College London, London, UK
[9]Department of Infectious and Parasitic Diseases, Faculdade de Medicina da Universidade de Sao Paulo, São Paulo, São Paulo, Brazil
[10]Facultad de Ciencias de la Salud, Universidad de Carabobo, Valencia, Venezuela
[11]Mexican Institute of Social Security, Mexico City, Mexico
[12]Instituto de Pesquisa Clínica Evandro Chagas, Fundação Oswaldo Cruz, Rio de Janeiro, RJ, Brazil
[13]Department of Child and Adolescent Health, University of the West Indies at Mona, Kingston, Jamaica
[14]Instituto Aggeu Magalhães, Fundação Oswaldo Cruz, Recife, Brazil
[15]Instituto de Medicina Tropical Alexander von Humboldt, Universidad Peruana Cayetano Heredia, Lima, Peru
[16]INSERM Centre d'Investigation Clinique 1424, Centre Hospitalier Universitaire de Pointe-à-Pitre, Guadeloupe, France
[17]Faculté de Médecine Hyacinthe Bastaraud, Université des Antilles et de la Guyane, Pointe-à-Pitre, Guadeloupe, France
[18]Department of Viroscience, Erasmus Universiteit Rotterdam, Rotterdam, Zuid-Holland, The Netherlands
[19]Universidade Federal de Minas Gerais, Belo Horizonte, MG, Brazil
[20]Department of Infectious Diseases and Microbiology, University of Pittsburgh, Pittsburgh, Pennsylvania, USA
[21]Universidad Industrial de Santander, Bucaramanga, Colombia
[22]Universidade de Pernambuco, Recife, Brazil
[23]Figueira National Institute for Women's, Children's and Adolescents Health, Oswaldo Cruz Foundation, Rio de Janeiro, Rio de Janeiro, Brazil
[24]Emerging Pathogens Institute, University of Florida, Gainesville, Florida, USA
[25]Centro Nacional de Enfermedades Tropicales, Santa Cruz de la Sierra, Santa Cruz de la Sierra, Bolivia
[26]Universidad Católica de Santiago de Guayaquil, Guayaquil, Guayas, Ecuador
[27]SOSECALI C., Ltda, Guayaquil, Ecuador
[28]Department of Medical Microbiology and Infection Prevention, University Medical Center Groningen, Groningen, The Netherlands

[29]Institute of Tropical Pathology and Public Health, Federal University of Goias, Goiânia, Brazil

[30]Department of Woman's and Child's Health, Università degli Studi di Padova, Padova, Italy

[31]Aix-Marseille Université Institut Universitaire de Technologie d'Aix-en-Provence, Aix-en-Provence, Provence-Alpes-Côte d'Azur, France

[32]Department of Epidemiology and Global Health, Umeå University, Umeå, Sweden

**Acknowledgements** The authors express their thanks to Tom Byrne for organising the references.

**Collaborators** Other members of the EC Zika Consortia Vertical Transmission Study Group are: Joshua Anzinger (Department of Microbiology, Virology Section, University of the West Indies, Mona Campus, Kingston, Jamaica; Global Virus Network, Baltimore, MD, USA), Heather Bailey (Institute for Global Health, University College London, UK) Valery EM Beau De Rochars (Emerging Pathogens Institute, University of Florida, Gainesville, Florida, United States of America), Luana Damasceno (Evandro Chagas National Institute of Infectious Diseases, Fundação Oswaldo Cruz, Rio de Janeiro, Brazil), Maria de Fatima Pessoa Militão De Albuquerque (Instituto Aggeu Magalhães, Fundação Oswaldo Cruz, Recife, Pernambuco, Brazil), Rafael F O França (Instituto Aggeu Magalhães, Fundação Oswaldo Cruz, Recife, Pernambuco, Brazil), Hugo Lopez Gatell Ramirez (Centro de Investigación sobre Enfermedades Infecciosas, Instituto Nacional de Salud Pública, Cuernavaca, Mexico), Adriana Gomez (Universidad Industrial de Santander, Bucaramanga, Colombia), Fabiana Gordon (Population Health Sciences, Bristol Medical School, University of Bristol, Bristol, United Kingdom), Maria G Guzman (Instituto de Medicina Tropical Pedro Kourí, La Havana, Cuba), Daniel Lang (Fondazione Penta Onlus, São Paulo, São Paulo, Brazil), Rigan Louis (Emerging Pathogens Institute, University of Florida, Gainesville, Florida, United States of America), Anyela Lozano (Universidad Industrial de Santander, Bucaramanga, Colombia), Eric Martinez (Instituto de Medicina Tropical Pedro Kourí, La Havana, Cuba), Philippe Mayaud (Department of Clinical Research, London School of Hygiene and Tropical Medicine, London, UK), Ricard Ortiz Serrano (Universidad Autónoma de Bucaramanga, Bucaramanga, Colombia), Silvia P Salgado (Instituto Nacional de Salud Pública e Investigación, Guyaquil, Ecuador), Aluisio Augusto Cotrim Segurado (Department of Infectious and Parasitic Diseases and Instituto de Medicina Tropical, Faculdade de Medicina da Universidade de Sao Paulo, Sao Paulo, Brazil), Zilton F M Vasconcelos (Instituto Fernandes Figueira – Fundação Oswaldo Cruz, Rio de Janeiro, Rio de Janeiro, Brazil), Isabelle Freire Tabosa Viana (Instituto Aggeu Magalhães, Fundação Oswaldo Cruz, Recife, Pernambuco, Brazil), Luis Angel Villar (Universidad Industrial de Santander, Bucaramanga, Colombia).

**Contributors** A steering group (CT, TJ, KDR, RAdAX, AEA, EBB and NA) was formed to prepare a Joint Statistical Analysis Plan pursuant to the terms of the three EC Zika Consortia funding, at the direction of the consortia PIs (CG, Xd-L and AW-S). After initial discussions the steering group delegated the task to AEA, EBB and NA. The SAP was conceptualised and drafted by AEA with EBB and NA. After several iterations, the draft was revised based on comments from CT, DB, AS-A, TJ, KDR, MP, DdBM-F, RAdAX, MCMM, AT, CG, Xd-L and AW-S. Comments were then solicited from TVBdA, VIA-S, SB, VB-A, PB, CDCC, WDS, JEGH, BH, MK, CMTM, MT, ETAM, URM, MEM, JGM, BR, PMSV, CSS and MDT and a revision was drafted by AEA, EBB and NA. The final draft was agreed by all authors.

**Funding** European Union Directorate-General for Research grant numbers 734 548 (ZIKAlliance), 734 584 (ZikaPLAN) and 734 857 (ZIKAction) Horizon 2020 programme of the European Commission. EBB was also supported by Wellcome Trust & the UK's Department for International Development (205377/Z/16/Z to LCR; https://wellcome.ac.uk/).

**Competing interests** All authors have completed the ICMJE uniform disclosure form. The following authors report grants from the European Commission (AEA, EBB, NA, DB, TJ, KDR, CT, CDCC, AS-A, JGM, CG, Xd-L, AW-S); EBB reports funding from by Wellcome Trust & the UK's Department for International Development; MT reports grants from Conselho Nacional de Desenvolvimento Cientifico e Tecnologico, Brazil and from Fundacao de Amparo a Pesquisa do EStado de MInas Gerais (FAPEMIG, Brazil), during the conduct of the study; MK has a patent on zika diagnostics pending.

**Patient consent for publication** Not required.

**Ethics approval** The following ethical committees / institutional review boards have approved the participating studies: Bolivia: El Tribunal de Etica del Colegio Médico de Santa Cruz -Tribunal Departamental de Etica y Deontología Médica (TEDM) (018/2017; 019/2017). Brazil: Comissão Nacional de Ética em Pesquisa (CONEP) (2.493.696/CAAE 67410117.3.1001.5262); Comitê de Ética em Pesquisa do Hospital Sofia Feldman (CEP/HSF) (2.549.675/CAAE 67410117.3.3001.5132); Comitê de Ética em Pesquisa da UFMG (COEPE-UFMG), Belo Horizonte (2.877.160/CAAE 67410117.3.2002.5149); Comitê de Ética em Pesquisa do Instituto Nacional de Infectologia Evandro Chagas (INI), Manguinhos - Rio de Janeiro; Comitê de Ética em Pesquisa da Faculdade de Medicina da Universidade de São Paulo (CEP-FMUSP) (2.195.948 /CAAE 63567917.6.0000.0065; 2.262.420/CAAE 74853517.0.0000.0065; 1.938.335/CAAE 54699416.0.0000.0065); Comitê de Ética em Pesquisa (CEP) do Instituto Aggeu Magalhães, Recife (2.777.436/CAAE 80243817.5.0000.5190; 2.737.404/CAAE 67410117.3.2001.5190); CEP Universidade Federal de Goiás, Goiânia (64534017.7.0000.5083); CEP Instituto Fernandes Figueira, Rio de Janeiro (52675616.0.0000.5269); CEP Instituto Aggeu Magalhães, Recife, Brazil (53240816.4.0000.5190) Colombia: Comité de Ética en Investigación Científica de la Universidad Industrial de Santander (CEINCI-UIS) (4110). Cuba: Comité de Ética de la Investigación del Instituto de Medicina Tropical "Pedro Kourí" (25–18; 59–16; 40–16). Ecuador: Comité de Ética del Hospital Clínica Kennedy, Guayaquil (HCK-CEISH-17-0026). Haiti: Ministère de la Santé Publique et de la Population, Comité National de Bioéthique (1718-7). Jamaica: University of the West Indies (Mona Campus) Research Ethics Committee (104-16/17). Mexico: Comité de Ética en Investigación y Comité de Investigación de la Comisión Nacional de Investigación Científica del IMSS (R-2016-785-076; R-2018-785-42); Comité de Ética en Investigación del Instituto Nacional de Salud Pública y Comité de Bioseguridad del Instituto Nacional de Salud Pública (17CEI00420160708/13CEI1700736; 18CI17007029/CI-620-2018; CB18–133; CI-369-2017; CB17–143). Peru: Comité Institucional de Ética en Investigación de la Universidad Peruana Cayetano Heredia (101582/766–22-17; 101582/R-234-25-18). UK: University College London Research Ethics Committee (3715/001). US: Institutional Review Board, University of Florida (IRB201601229) Venezuela: Comisión de Bioética y Bioseguridad, Universidad de Carabobo (CPBB-UC) (CPBBUC-020-2017-DIC-AS; CPBBUC-008-2018-DIC-SB.). Findings will be presented at international conferences and published in peer-reviewed open access journals, and discussed with local public health officials and representatives of the national Ministries of Health, Pan American Health Organization, and World Health Organization involved with ZIKV prevention and control activities.

**Provenance and peer review** Not commissioned; externally peer reviewed.

**ORCID iDs**
Antoni Soriano-Arandes http://orcid.org/0000-0001-9613-7228
Claire Thorne http://orcid.org/0000-0003-0389-1956
Adriana Tami http://orcid.org/0000-0002-1918-9144

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
