## [Reviewer comments · BMJ Open]

ARTICLE DETAILS

TITLE (PROVISIONAL)	Zika virus infection in pregnancy: A protocol for the joint analysis of the prospective cohort studies of the ZIKAlliance, ZikaPLAN and ZIKAction consortia
AUTHORS	Ades, A E; Brickley, Elizabeth B.; Alexander, Neal; Brown, David; Jaenisch, Thomas; Miranda-Filho, Demócrito de Barros; Pohl, Moritz; Rosenberger, Kerstin Daniela; Soriano-Arandes, Antoni; Thorne, Claire; Arraes de Alencar Ximenes, Ricardo; Velho Barreto de Araújo, Thalia; Avelino-Silva, Vivian; Bethencourt, Sarah; Borja-Aburto, Victor; Brasil, Patrícia; Christie, C. D. C.; DE SOUZA, Wayner; GOTUZZO H., Jose Eduardo; Hoen, Bruno; Koopmans, M; Turchi Martelli, Celina Maria; Teixeira, Mauro; Marques, Ernesto; Miranda Montoya, Maria Consuelo; Ramos Montarroyos, Ulisses; Moreira, Maria; Morris, J; Rockx, Barry; SABA VILLARROEL, Paola Mariela; SORIA SEGARRA, Carmen; Tami, Adriana; Turchi, Marília Dalva; Giaquinto, Carlo; de-Lamballerie, Xavier; Wilder-Smith, Annelies

VERSION 1 – REVIEW

REVIEWER	David A. Schwartz, MD, MS Hyg, FCAP Medical College of Georgia Augusta University USA
REVIEW RETURNED	05-Dec-2019

GENERAL COMMENTS	The panel of highly qualified authors of this well-organized and clearly-written manuscript have designed a joint protocol for the pooled statistical analysis of the incidence of adverse outcomes of Zika virus infection occurring during pregnancy among three separately funded cohort studies - ZikaPLAN, ZIKAlliance, and ZIKAction, including comparisons with non-Zika virus infected women. It will also investigate markers of vertical Zika virus transmission as well as the effects of risk covariables. This is an ambitious project involving the collection, processing, pooling and analysis of data from 20 regional coordinating centers in 11 countries throughout Latin America and the Caribbean. The results of this proposed analysis will be of tremendous benefit to not only increasing knowledge of the effects of Zika virus during pregnancy, but can potentially provide a model for the pooled analysis of studies of other important congenitally-transmitted maternal-fetal infections including dengue, Ebola and Lassa fever viruses.
--

REVIEWER	Samantha M. Olson, MPH G2S Corporation; National Center on Birth Defects and Developmental Disabilities,
-----------------	--

	Centers for Disease Control and Prevention, United States of America
REVIEW RETURNED	26-Feb-2020

GENERAL COMMENTS	Review: Zika virus infection in pregnancy: a protocol for the joint analysis of the prospective cohort studies of the ZIKAlliance, ZikaPLAN, and ZIKAction consortia This protocol aims to elucidate remaining questions regarding maternal Zika virus infection during pregnancy. A range of estimates of congenital infection has been presented from various cohorts and studies. This protocol could provide a valuable addition to the literature by estimating the risk of congenital infection and other potential maternal risk factors such as trimester of infection and presence or absence of symptoms, with the strength of using unified definitions/protocols across 20 centres in 11 countries. However, the main concern is the protocol's lack of pre-specifying exposures, outcomes, and planned analyses. The authors state that standardizing definitions of outcomes is a main strength of the protocol, yet do not present the proposed definitions in full. Standardizing definitions in advance or including more detail on processes by which each will be determined is needed to justify pre-publishing the protocol. Additionally, since a protocol with similar objectives has been published aiming to include these 20 centres, it is important to explain how this builds upon the existing protocol and establish how this protocol advances the literature. Main comments/concerns:  • The protocol should pre-specify exposure/outcome definitions, or further explain and support the decision to determine the case definitions for maternal infection in pregnancy (MIP) and ZIKV-related congenital outcomes at a later time by the Joint Diagnostics Group and Joint Endpoints Review Group. Who will be a part of this review committee, how will decisions be made, and when? Will the two review groups have access to exposure/outcome data when formulating standardized definitions, or will the definitions be based upon protocols alone? Would these committees be blinded to all or some of the data in making decisions? If the review groups will have access to data, please indicate if/how the data will be used in determining exposure/outcome definitions. • Please, elaborate on how this protocol's objectives are different than the Wilder-Smith, et al. protocol. Suggest adding an additional paragraph after page 7, line 14-16 to briefly summarize this protocol and then clearly delineate how this protocol differs. This is critical to explain how this protocol builds on this other meta-analysis plan to justify the publication of the current protocol. • Analytic plans for assessing the heterogeneity between centres should be explained to determine whether pooling across these studies are supported by the data. • The protocol would improve if authors could add additional details about the data from the centres in Supplementary Table S1 including...  o List of all studies included in each consortia including where to find each protocol. o Time period of enrollment for each cohort. This is required for BMJ Open Protocols and will help readers identify if the cohort
--

	enrollment overlapped with epidemic peaks in each of their respective countries.  o Expected sample sizes o Include specimen types for all testing being conducted and are there any additional studies where TORCH or testing for other infections are conducted? o Harmonize each section so it is written in a similar way across studies making it easier to identify similarities and differences between them.  • Additional detail around who is included in the control group for each of these consortia would be helpful. Are the control groups similar and comparable across these studies? If not, how will the analysis account for these differences. Additionally, if an infant tests positive and the mom tests negative, this case would not be included in the unexposed control group due to the infant testing? If this is correct, please clarify throughout protocol (specifically page 9, line 38; it would be helpful to add for “no evidence of MIP” that this is referring to all maternal and infant testing being negative). • Please, use consistent terms such as “congenital Zika syndrome” (e.g., page 5, line 33) or “vertical transmission” instead of “trans-placental transmission (e.g., page 7, line 8) throughout the manuscript. Additionally, clearly distinguish the meaning of “marker” (e.g., page 7, line 32); it is unclear if this refers to laboratory testing, clinical findings, or both? Additional line by line edits (referenced pages specified in originally submitted manuscript):  • Page 3  o Line 26: Previous flavivirus infection is mentioned as a key element to investigate in association with congenital infection; however, there are limited details on how this will be done or what data are collected related to this in the protocol. Please, provide more information or remove this from the abstract and main objectives if details are not available. Could mention as a potential analysis if enough data are available as an alternative. • Page 5  o Line 29: Could more detail be provided around how the pooled analysis will be conducted? Will an individual-level analysis (similar to the individual participant data meta-analysis in the previously published protocol) not be conducted until the completion of other analyses (referenced on page 13, line 17)? o Line 49: Change “adverse outcomes at rates of up to 15%, with higher rates in the first trimester.” to “adverse outcomes at rates of up to 15% in the first trimester.” A birth defects rate of 15% is only among confirmed first trimester infections not among all confirmed ZIKV infected pregnancies. This is necessary to edit to not misrepresent the estimate since the estimate among confirmed and probable pregnancies, in these cohorts, is much lower. o Line 56 (and on to first line of next page): Suggest modifying “was recognized retrospectively...” For the majority these infections were recognized during pregnancy and followed prospectively through longitudinal surveillance. It is possible that if an infant tested positive they could be included in the registry, but the infant would not be included based solely on the presence of an adverse outcome. There is another US-based system that tracks birth defects, Zika Birth Defects Surveillance, but this is not a registry and is not captured in any of the studies cited and would not provide a risk estimate of birth defects but would provide the prevalence of birth defects during that time period. Thus, suggest
--	---

	modifying sentence to be more indicative of studies cited unless there are other Zika registries that could be cited here.  • Page 8  o Line 34: Suggest providing a brief statement to introduce the use of groups mentioned in Table 1 such as, “Six categories of joint congenital infection status and maternal infection status (A-F) are defined in Table 1.” • Page 9  o Line 8: Should this be “combined” instead of “comprised”? • Page 13  o Line 3: Can you elaborate on your priors for the proposed Bayesian analysis? • Page 15:  o Line 8: Can you elaborate on “whatever form of analysis is adopted”? All analyses might not be defined yet, but more detail would improve this protocol. • Page 16  o Line 53: Could you define all the acronyms mentioned? That may make this section more meaningful. • Page 24  o Line 24: In the “Most likely trimester of onset of maternal infection” section for the last item in this section’s hierarchy (i.e., “tests of recent infection”) is this referring to NAAT or other tests? If it is NAAT, this seems to contradict the first point of the hierarchy. Please, define which tests this is referring to? For the first point it might also be helpful to define the average duration of viremia or cite? Is this 14 days from citation 32, Paz-Bailey G, et al.? • Page 25  o Line 34: Might consider adding other “evidence of congenital infection,” specifically birth defects such as other brain abnormalities and eye abnormalities. If those clinical data are available for interpretation.
--	---

REVIEWER	Debora Diniz International Planned Parenthood Federation/Western Hemisphere Region (IPPF/WHR), USA.
REVIEW RETURNED	26-Feb-2020

GENERAL COMMENTS	I have a few minor questions though. When was the study started? I understood that data collection is ongoing, but not sure for how long that has been happening for and whether there will be an endpoint. Is there consideration for sample size? or consideration for target number of participants for statistical power? How many participants are there so far? Wouldn't the potential impact of loss to follow up be an issue that deserves separate consideration in such large cohort study of pregnancy outcomes? The protocol seems adequate for what it is set to accomplish. This large, multicenter prospective cohort studies in Latin America and the Caribbean on pregnancy outcomes related to Zika infection is relevant and needed research. Collaboration among these 20 centers in 11 different countries has the potential to advance the scientific knowledge about Zika and CZS in a significant manner for the region. It must not be overlooked that the most affected by Zika are the most vulnerable women and children, thus particular attention to research ethics that accounts for the local realities and benefit sharing should always be kept in mind of all of those involved in the study.
---

REVIEWER	Jovana Alexandra Ocampo Canas Andes University
REVIEW RETURNED	28-Feb-2020

GENERAL COMMENTS	Thanks so much for this innovative protocol! My understanding of the main goal of this paper is to present a protocol for analyzing a pool of data have been gathered and that they are been still collecting in three research Consortia. However, when I read the objectives section, the main goal is not specified, which could confuse the audience who reads the paper. Line 35: please define “MIP” meaning. I know you define in the supplementary section. However, it is important define it from the very beginning in order to have a much easier understanding of the paper. It is important to mention if you got or not the women’s consent. If so, how did you get it? As you might know, the nature of this study requires that women have a consent.
---

VERSION 1 – AUTHOR RESPONSE

Reviewer: 1

Reviewer Name: David A. Schwartz, MD, MS Hyg, FCAP Institution and Country:
Medical College of Georgia
Augusta University
USA

Please state any competing interests or state ‘None declared’: None declared

The panel of highly qualified authors of this well-organized and clearly-written manuscript have designed a joint protocol for the pooled statistical analysis of the incidence of adverse outcomes of Zika virus infection occurring during pregnancy among three separately funded cohort studies - ZikaPLAN, ZIKAlliance, and ZIKAction, including comparisons with non-Zika virus infected women. It will also investigate markers of vertical Zika virus transmission as well as the effects of risk covariables. This is an ambitious project involving the collection, processing, pooling and analysis of data from 20 regional coordinating centers in 11 countries throughout Latin America and the Caribbean. The results of this proposed analysis will be of tremendous benefit to not only increasing knowledge of the effects of Zika virus during pregnancy, but can potentially provide a model for the pooled analysis of studies of other important congenitally-transmitted maternal-fetal infections including dengue, Ebola and Lassa fever viruses.

Thank you for these kind words. But we claim no originality for this protocol: it is modelled closely on classic studies such as Miller (1982) on rubella and other papers cited in our protocol: Stagno (1982) and Townsend (2013) on cytomegalovirus; Dunn(1994) and Mandelbrot (1996) on HIV; Gras (2001) on toxoplasmosis; among many others.

The revised draft emphasises that we are following the pattern set out in these papers.

Reviewer: 2

Reviewer Name: Samantha M. Olson, MPH
Institution and Country:
G2S Corporation;
National Center on Birth Defects and Developmental Disabilities, Centers for Disease Control and Prevention, United States of America Please state any competing interests or state ‘None declared’:
None declared.

This protocol aims to elucidate remaining questions regarding maternal Zika virus infection during pregnancy. A range of estimates of congenital infection has been presented from various cohorts and studies. This protocol could provide a valuable addition to the literature by estimating the risk of

congenital infection and other potential maternal risk factors such as trimester of infection and presence or absence of symptoms, with the strength of using unified definitions/protocols across 20 centres in 11 countries.

Thank you for these positive comments.

However, the main concern is the protocol's lack of pre-specifying exposures, outcomes, and planned analyses. The authors state that standardizing definitions of outcomes is a main strength of the protocol, yet do not present the proposed definitions in full. Standardizing definitions in advance or including more detail on processes by which each will be determined is needed to justify pre-publishing the protocol.

We will address this below (under "Main comments/concerns").

Additionally, since a protocol with similar objectives has been published aiming to include these 20 centres, it is important to explain how this builds upon the existing protocol and establish how this protocol advances the literature.

We agree that publication is only warranted if the protocol is both scientifically appropriate and clearly different from the existing protocol of Wilder-Smith *et al* (2019). In fact, our protocol does not "build on" the latter but instead it proposes an alternative approach using basic principles which have been tried and tested in classic studies of vertical transmission (see response to reviewer 1). These methods are, of course, adapted to deal with the specific diagnostic problems with Zika virus. regarding detection of both maternal infection in pregnancy and congenital infection. This is achieved by generating a series of estimates of the target parameters, based on different numerators and denominators. For example, estimates of the vertical transmission rate might be based on "Confirmed MIP" alone, or on "Confirmed MIP" and "Suspected MIP" together.

Main comments/concerns:

- The protocol should pre-specify exposure/outcome definitions, or further explain and support the decision to determine the case definitions for maternal infection in pregnancy (MIP) and ZIKV-related congenital outcomes at a later time by the Joint Diagnostics Group and Joint Endpoints Review Group.

It would be premature to commit ourselves at this point to specific case definitions of maternal infection in pregnancy (MIP) and ZIKV-related congenital outcomes, for several reasons:

1. We anticipate that each consortium will wish to publish analyses of its own data before agreeing to share data for a joint analysis. The joint analysis is therefore some time off.
2. In the intervening period, knowledge of existing diagnostics will increase and new assays will be developed and applied to our stored samples.
3. We can also expect improved understanding of which fetal, newborn, and pediatric outcomes are ZIKV-related. Supplementary Table 1 gives a clear indication of our current line of thought, but we must be allowed to take advantage of any new knowledge that will become available before the joint analysis starts

In revision we have noted that diagnostic categories and clinical definitions will be based on the best information available at the time of analysis. Please note that the Wilder-Smith protocol, which the reviewer refers to below, also classifies maternal infection as "confirmed", "probable", "unlikely", etc (See WS Table 1), without specifying the precise definitions of these terms.

Who will be a part of this review committee, how will decisions be made, and when?

As the final analysis is some way in the future, we are reluctant to give names. We have revised the text to say that the Diagnostics group will consist of experts in virology and immunology, and the Endpoint group will consist of paediatricians with first-hand experience of congenital ZIKV. We have also added that the names and qualifications of our experts and the rationale and process for their recruitment will be published along with the data analysis.

... how will decisions be made, and when? Will the two review groups have access to exposure/outcome data when formulating standardized definitions, or will the definitions be based upon protocols alone? Would these committees be blinded to all or some of the data in making decisions? If the review groups will have access to data, please indicate if/how the data will be used in determining exposure/outcome definitions.

These questions about whether the review groups will be blinded are important ones which we have debated: we thank the reviewer for raising this. Ideally, the experts working on diagnostic definitions would be blind to clinical outcomes, and vice versa. Our original intention was to recruit experts from within the consortia, and several have agreed to act in this capacity. However, the fact that each consortium will have already thoroughly examined their own data will make it impossible to operationalise blinding. The alternative is to recruit experts from outside the three consortia, but we have no idea whether this would be feasible and it would not be practical to try to arrange this so far in advance.

We must therefore fall back on recruitment of experts from within the three consortia. The revised text states that the expert groups will not be blinded to each other's data, but will draw up explicit criteria for exposure and outcome definitions which can be objectively applied. The data will be available to other researchers who may wish to experiment with alternative definitions.

- *Please, elaborate on how this protocol's objectives are different than the Wilder-Smith, et al. protocol. Suggest adding an additional paragraph after page 7, line 14-16 to briefly summarize this protocol and then clearly delineate how this protocol differs. This is critical to explain how this protocol builds on this other meta-analysis plan to justify the publication of the current protocol.*

Professor Wilder-Smith *et al.* propose a diverse array of advanced statistical methods, including lasso regression, G-computation, marginal structural models, multi-variate normal random effects meta-analysis, and many others that are not fully specified. By contrast, our protocol was designed from the outset to be as simple as possible and consistent with a literature on congenital infection going back almost 40 years.

It is important to recognise that although our objectives are the same as Wilder-Smith's objectives (1) and (2), the two studies have different scopes and methodologies; so much so that if given identical bodies of data, each would produce different results - perhaps markedly different. We focus on three fundamental issues:

1. *Study inclusion.* The Wilder-Smith analysis will include not only prospective cohort studies but also case-cohort studies and data from surveillance. The need for prospective ascertainment of women's infection status is not mentioned. There is therefore a real risk that women will be included whose ZIKV infection in pregnancy was only detected as a result of adverse fetal or newborn outcomes, resulting in over-estimation of the risk of adverse outcomes. (See response to reviewer's comments on Page 5 line 56)
2. *Definition of congenital infection.* While congenital infection is an *outcome* in our protocol and the poor diagnostic sensitivity of laboratory markers is recognised, in Wilder-Smith it is treated both as an outcome and as an exposure, which is to be defined on clinical and radiological criteria (See WS Table 1 and footnotes). Our protocol will use laboratory criteria, such as PCR and IgM, to define congenital infection. This is the standard approach, used in all the studies cited in response to Reviewer 1.
3. *Inaccuracy of diagnosis of maternal infection.* Our protocol will classify maternal infection in pregnancy as "definite", "probable", "no evidence of infection in pregnancy", "no infection in pregnancy" (as defined by an Expert Diagnostics Group prior to the analysis) and will simply tabulate outcomes in each category, to generate a range of estimates. Wilder-Smith proposes to run regressions with the absolute and/or relative risks of adverse outcomes as the dependent variable, and with estimates of test sensitivity and specificity as study-level covariates. They do not specify exactly how unbiased estimates of absolute and relative risks will be extracted from this analysis. Also, the probability that a woman's infection status is incorrectly classified does not depend on test sensitivity and specificity alone: it may be far more dependent on the testing protocol, degree of compliance with the protocol, and particularly on the incidence and prevalence of ZIKV and of other viruses such as dengue that can cause cross-reactions.

As suggested by the reviewer, we have expanded our comments briefly on the Wilder-Smith protocol to include these three points. We do not believe that detailed comparison is warranted in the text, because it will be clear that our approach is based the classic studies of congenital infection, while Wilder-Smith takes several rather novel approaches.

- *Analytic plans for assessing the heterogeneity between centres should be explained to determine whether pooling across these studies are supported by the data.*

Our understanding of the literature so far leads us to expect a considerable degree of between-centre variation in risks of adverse outcomes. In revision, we have added that we will report ranges,

between-centre standard deviations, mean and predictive effects and the credible intervals. We do not accept that the degree of heterogeneity has any bearing on whether pooling should be done: there is nothing in the theory of random effect models that supports this notion. We are aware that random effect means have no true population interpretation as they depend on which centres are included. However, pooled means over such a large number of centres will be of interest.

- *The protocol would improve if authors could add additional details about the data from the centres in Supplementary Table S1 including...*

- *List of all studies included in each consortia including where to find each protocol.*
We have added references to consortia protocols and methodology papers on ZIKV in pregnancy, where these have been published. These appear where the consortia are introduced on page 5.

Regarding the reviewer's request for protocols for "all studies included in each consortia", each consortium embraces an exceptionally wide range of multi-disciplinary activities, from studies of the viral genome to in vivo studies on laboratory models. It would not be appropriate to include references in this paper.

- *Time period of enrollment for each cohort. This is required for BMJ Open Protocols and will help readers identify if the cohort enrollment overlapped with epidemic peaks in each of their respective countries.*

Enrollment started at different dates for each centre. ZIKAlliance and ZIKAction centres began recruitment at a time when the ZIKV outbreaks were already tapering off. For ZikaPLAN recruitment started in December 2015. Similarly, recruitment continues in some centres, but not in others. This information on consortium start dates has been added to the revised text. We do not believe that further detail would be helpful to readers.

Expected sample sizes

Regarding samples size calculations: when this study began very little was known about the consequences of maternal ZIKV infection in pregnancy. We had no idea of what the vertical transmission or adverse outcome rates might be, nor – more importantly – what the incidence of maternal infection would be. Formal sample size calculations would not have been appropriate. Upper limits to the number of infected women to be recruited were discussed, but these were never reached. We have added a brief explanation to the Study Design section. Regarding the numbers expected now (at the time of writing), we can report that ZikaPLAN has evaluated over 700 children born to pregnant women with confirmed infection. ZIKAction has recruited several hundred women and ZIKAlliance over 4500. A summary statement has been added to the revised text

- *Include specimen types for all testing being conducted and are there any additional studies where TORCH or testing for other infections are conducted?*

Bearing in mind that the formal definitions remain to be elaborated by the expert diagnostics group, and that testing algorithms for stored samples are still being elaborated, we have provided all the necessary details in Supplementary Table 1, including in particular details of sample types.

Regarding "additional studies", we are not able to document all the sub-studies that might be being conducted in the collaborating centres

Harmonize each section so it is written in a similar way across studies making it easier to identify similarities and differences between them.

We do not completely understand this comment. The key differences between consortia protocols are described in the Study Design section.

- *Additional detail around who is included in the control group for each of these consortia would be helpful. Are the control groups similar and comparable across these studies? If not, how will the analysis account for these differences.*

We have added a brief sentence in the Study Design section to ZikaPLAN to clarify that all three consortia included a comparable control group. The existing text notes that in ZikaPLAN only women with rash in pregnancy were recruited, and that stratification will be used to handle this in the data analysis.

Additionally, if an infant tests positive and the mom tests negative, this case would not be included in the unexposed control group due to the infant testing? If this is correct, please clarify throughout

protocol (specifically page 9, line 38; it would be helpful to add for “no evidence of MIP” that this is referring to all maternal and infant testing being negative).

If the mother tests negative, meaning PCR-ve and no serological evidence of infection in pregnancy, the mother would be classified as “no evidence of MIP”. This classification will not be changed whether the infant tests positive or negative for markers of congenital infection. In revision we have strengthened the text by noting that this is to preserve the principle of prospective ascertainment. As noted elsewhere in the text, if cases of congenital infection occur in the “No Evidence of MIP” group, this will be taken as evidence of the lack of overall program sensitivity of the maternal testing process, and/or lack of compliance.

A somewhat different situation *may* exist if the infant or the mother test IgG negative at delivery, AND there are no positive markers of congenital infection. If our Expert Diagnostics Group regards this as evidence that the mother *could not have been infected in pregnancy*, then the mother would be classified in the “no infection in pregnancy” group.

- Please, use consistent terms such as “congenital Zika syndrome” (e.g., page 5, line 33) or “vertical transmission” instead of “trans-placental transmission (e.g., page 7, line 8) throughout the manuscript.

Agreed, we have made the necessary revisions.

Additionally, clearly distinguish the meaning of “marker” (e.g., page 7, line 32); it is unclear if this refers to laboratory testing, clinical findings, or both?

Thank you for pointing this out. In the revised text we have allowed for *both* laboratory markers and clinical markers (ie CZS) of congenital infection. This is clarified throughout. We have made the necessary amendments to Table 3 and to the supplementary tables. This has led us to make an important clarification: that the Endpoint Review Group will aim to produce a definition of CZS that is as close as possible to 100% specific for congenital ZIKV infection. Our view is that – in the context of women exposed to ZIKV outbreaks, such a definition is entirely feasible, especially if TORCH can be ruled out. This is noted in Supplementary Table S2.

Additional line by line edits (referenced pages specified in originally submitted manuscript):

- Page 3

- o Line 26: *Previous flavivirus infection is mentioned as a key element to investigate in association with congenital infection; however, there are limited details on how this will be done or what data are collected related to this in the protocol. Please, provide more information or remove this from the abstract and main objectives if details are not available. Could mention as a potential analysis if enough data are available as an alternative.*

Agreed: the revised text notes that the definition of previous flavivirus infection will be another task for the Diagnostics Group

- Page 5

- o Line 29: *Could more detail be provided around how the pooled analysis will be conducted? Will an individual-level analysis (similar to the individual participant data meta-analysis in the previously published protocol) not be conducted until the completion of other analyses (referenced on page 13, line 17)?*

As stated in the current draft, an IPD analysis will be conducted depending on the results of the “two-stage” analysis. However, it will not be “similar to the IPD meta-analysis” in Wilder-Smith *et al.* owing to the different way outcomes are defined and the different approach to imperfect diagnostics for maternal infection (see previous comments on differences between the current approach and Wilder-Smith).

- o Line 49: *Change “adverse outcomes at rates of up to 15%, with higher rates in the first trimester.” to “adverse outcomes at rates of up to 15% in the first trimester.” A birth defects rate of 15% is only among confirmed first trimester infections not among all confirmed ZIKV infected pregnancies. This is necessary to edit to not misrepresent the estimate since the estimate among confirmed and probable pregnancies, in these cohorts, is much lower.*

Agreed. The wording was misleading and we have made this revision

- o Line 56 (and on to first line of next page): *Suggest modifying “was recognized retrospectively...” For the majority these infections were recognized during pregnancy and followed prospectively through longitudinal surveillance. It is possible that if an infant tested positive they could be included in the registry, but the infant would not be included based solely on the presence of an*

adverse outcome. There is another US-based system that tracks birth defects, Zika Birth Defects Surveillance, but this is not a registry and is not captured in any of the studies cited and would not provide a risk estimate of birth defects but would provide the prevalence of birth defects during that time period. Thus, suggest modifying sentence to be more indicative of studies cited unless there are other Zika registries that could be cited here.

We accept that most of the infections were recognised during pregnancy, and that an “infant would not be included based solely on the presence of an adverse outcome”. But it is still the case that, if a mother who had not been tested delivered an infant with a potentially ZIKV-related anomaly, then the mother would be likely to be tested for ZIKV, and this would count as retrospective ascertainment. For example, Reynolds (*MMWR Morb Mortal Wkly Rep.* 2017 Apr 7; 66(13): 366–373) reports “Pregnant women with Zika virus exposure and prenatally detected fetal abnormalities or infants with birth defects might be more likely to be tested for Zika virus infection”.

Similarly, Honein (*JAMA.* 2017;317(1):59-68. doi:10.1001/jama.2016.19006) “Pregnant women with a history of possible exposure to Zika virus through travel or sex might have been more likely to be tested for Zika virus infection if fetal abnormalities were detected prenatally”.

The revised text refers to this earlier work

- Page 8

- o Line 34: Suggest providing a brief statement to introduce the use of groups mentioned in Table 1 such as, “Six categories of joint congenital infection status and maternal infection status (A-F) are defined in Table 1.”

Agreed: we have revised accordingly

- Page 9

- o Line 8: Should this be “combined” instead of “comprised”?

We cannot locate the this on Page 9 line 8. The word “comprising” is used in the last paragraph, and seems to be used correctly

- Page 13

- o Line 3: Can you elaborate on your priors for the proposed Bayesian analysis?

The existing text specifies “vague priors” and refers to Beta, Dirichlet, and Normal distributions. For these distributions there are fairly standard vague priors: Beta(1,1), Dirichlet(1,1,1), Normal(0,100²)

- Page 15:

- o Line 8: Can you elaborate on “whatever form of analysis is adopted”? All analyses might not be defined yet, but more detail would improve this protocol.

We added this sentence only to signal that, in the event that we devised a superior form of analysis, the prior publication of this protocol would not prevent us from implementing it. It would not be appropriate here to launch into a discussion of alternative statistical approaches, which would be no more than speculative.

- Page 16

- o Line 53: Could you define all the acronyms mentioned? That may make this section more meaningful.

These are the initials of the authors, which should be clear from the heading “Author contributions”. We believe this is standard practice

- Page 24

- o Line 24: In the “Most likely trimester of onset of maternal infection” section for the last item in this section’s hierarchy (i.e., “tests of recent infection”) is this referring to NAAT or other tests? If it is NAAT, this seems to contradict the first point of the hierarchy. Please, define which tests this is referring to? For the first point it might also be helpful to define the average duration of viremia or cite? Is this 14 days from citation 32, Paz-Bailey G, et al.?

“Tests of recent infection” refers to IgG3, IgM, IgG avidity. We have clarified by adding the word “serological”

- Page 25

- o Line 34: Might consider adding other “evidence of congenital infection,” specifically birth defects such as other brain abnormalities and eye abnormalities. If those clinical data are available for interpretation.

We have revised this table entirely, accepting this suggestion, but also sharpening the distinction between “CZS”, a marker of congenital ZIKV infection which is provisionally assumed to be virtually 100% specific, and other evidence for congenital ZIKV infection which might be indicative but not definitive.

Reviewer: 3

Reviewer Name: Debora Diniz

Institution and Country: International Planned Parenthood Federation/Western Hemisphere Region (IPPF/WHR), USA.

When was the study started? I understood that data collection is ongoing, but not sure for how long that has been happening for and whether there will be an endpoint.

Please see response to Reviewer 2

Is there consideration for sample size? or consideration for target number of participants for statistical power?

Please see response to Reviewer 2

How many participants are there so far?

Please see response to Reviewer 2

Wouldn't the potential impact of loss to follow up be an issue that deserves separate consideration in such large cohort study of pregnancy outcomes?

Loss to follow-up is a concern. We have revised the text to note that we will use the numbers of livebirths as the denominator when reporting incidence of adverse outcomes in childhood.

The protocol seems adequate for what it is set to accomplish. This large, multicenter prospective cohort studies in Latin America and the Caribbean on pregnancy outcomes related to Zika infection is relevant and needed research. Collaboration among these 20 centers in 11 different countries has the potential to advance the scientific knowledge about Zika and CZS in a significant manner for the region. It must not be overlooked that the most affected by Zika are the most vulnerable women and children, thus particular attention to research ethics that accounts for the local realities and benefit sharing should always be kept in mind of all of those involved in the study.

All consortia protocols required informed consent, and all were approved by local review boards.

Reviewer: 4

Reviewer Name: Jovana Alexandra Ocampo Canas Institution and Country: Andes University

Thanks so much for this innovative protocol!

Thank you for these kind words!

My understanding of the main goal of this paper is to present a protocol for analyzing a pool of data have been gathered and that they are been still collecting in three research Consortia. However, when I read the objectives section, the main goal is not specified, which could confuse the audience who reads the paper.

The objectives are stated concisely in a separate section “OBJECTIVES OF THE JOINT ANALYSIS”. Each objective has a section in the STATISTICAL ANALYSIS section, which explains what will be done to achieve each objective

Line 35: please define “MIP” meaning. I know you define in the supplementary section. However, it is important define it from the very beginning in order to have a much easier understanding of the paper. The first use of the abbreviation MIP is in the Study Design section, where it is defined.

It is important to mention if you got or not the women´s consent. If so, how did you get it? As you might know, the nature of this study requires that women have a consent.

Yes, informed written consent was obtained in every centre

VERSION 2 – REVIEW

REVIEWER	Debora Diniz International Planned Parenthood Federation/Western Hemisphere Region (IPPF/WHR), USA.
REVIEW RETURNED	06-Aug-2020

GENERAL COMMENTS	Prior suggestions were addressed adequately
---

REVIEWER	Laura Divens Zambrano Centers for Disease Control and Prevention, USA
REVIEW RETURNED	04-Sep-2020

GENERAL COMMENTS	This manuscript is well-written and clearly outlines the methods that will be undertaken to pool results from consortia overseeing the implementation of multiple prospective cohort studies examining the impact of Zika infection during pregnancy. This body of work will be of critical importance in understanding adverse outcomes associated with congenital Zika infection, and represents the largest effort to date to combine prospective study data to further characterize these events. There are a few comments below for the authors to consider, with minor revisions requested. Introduction (page 5) Line 20: It could be helpful to briefly outline distinctions/differences between these three consortia in terms of their foci and approaches. Line 47: Consider re-ordering the consortia so that they are listed in a consistent order when mentioned together throughout the document. Line 54: "a higher proportion of adverse outcomes" (these are proportions, not rates; also, consider using language similar to this: "with higher incidence of Zika-associated birth defects" or "higher risk of Zika-associated birth defects in the first trimester." Page 6 Line 31: Should be noted (or expressed as a limitation) that IgM is not confirmatory. Line 36: Which tests? Serologic tests generally have limited specificity (particularly in the context of flavivirus cross-reactivity. But high specificity/sensitivity are characteristics of NAATs. Page 7 Line 39 (Objective 2): Specify what is meant by deliveries to women? Not all products of conception will be liveborn, so it might be better to specify "among fetuses and liveborn infants" instead of "deliveries" (or something along those lines) Page 8 Line 3: Please specify how infection status during pregnancy will be ascertained. (It may differ between studies – if so, please mention this.) Page 9 Line 20: Can you specify laboratory markers of vertical infection that will generally be used? Line 51: Generally, serologic tests (after ruling out DENV cross-reactivity) would indicate probable infection, although timing of infection cannot be inferred (and, as you aptly state, would not necessarily indicate an infection that occurred during pregnancy). Because of implications of "confirmed" infection (implying NAAT), I would suggest re-wording. Something along the lines of, "These
---

	tests therefore indicate probable (or presumptive) Zika of flavivirus infection, although timing of infection with regard to pregnancy cannot be ascertained.” Page 10 Line 6: Suggest removing the sentence, “An IgG negative response in the woman shortly after delivery can be taken as suggestive of No MIP.” This does not rule out infections later in pregnancy, from which an IgG response may not have yet mounted. Pages 10 Lines 12-13: Suggest maintaining adherence to laboratory classifications established by WHO or CDC case definitions, particularly given the cross-cutting nature of these consortia. Confirmed is generally reserved for NAAT or IgM with positive ZIKV PRNT; probable is reserved for ZIKV IgM assays with either a negative DENV IgM or where no PRNT was performed. Suspected indicates clinical criteria and epidemiologic linkage criteria fulfilled. Please consider following this classification scheme throughout. Page 11 Line 41: (General comment) Make sure that means of defining microcephaly are standardized across consortia, if possible. Page 15 (For Discussion section): It’s important to note that the timeframe for data collection for two of the consortia (ZIKAction and ZIKAlliance) began after the peak of the Zika epidemic, as included studies were initiated in mid-2017 while peak transmission in the Americas occurred in mid-2016. This may affect sample size relative to ascertainment of laboratory confirmation of cases and subsequent outcomes. General comment for tables: Consider revising tables to reflect standard laboratory-based case definitions (confirmed, probable, and suspected cases), as described previously. General comment for supplementary materials: Information on serology testing is a little vague. Could you specify any anticipated follow-up testing to rule out serologic cross-reactivity with DENV (e.g., IgM ELISA or PRNT).
--	---

VERSION 2 – AUTHOR RESPONSE

Reviewer: 3

Reviewer Name: Debora Diniz

Institution and Country: International Planned Parenthood Federation/Western Hemisphere Region (IPPF/WHR), USA.

Please state any competing interests or state ‘None declared’: None declared

Please leave your comments for the authors below Prior suggestions were addressed adequately

Reviewer: 5

Reviewer Name: Laura Divens Zambrano

Institution and Country: Centers for Disease Control and Prevention, USA Please state any competing interests or state ‘None declared’: None declared

This manuscript is well-written and clearly outlines the methods that will be undertaken to pool results from consortia overseeing the implementation of multiple prospective cohort studies examining the impact of Zika infection during pregnancy. This body of work will be of critical importance in

understanding adverse outcomes associated with congenital Zika infection, and represents the largest effort to date to combine prospective study data to further characterize these events. There are a few comments below for the authors to consider, with minor revisions requested.

Introduction (page 5)

Line 20: It could be helpful to briefly outline distinctions/differences between these three consortia in terms of their foci and approaches.

Each of the three consortia encompasses a very wide range of approaches, spanning detailed virological and immunological studies, to animal studies, and epidemic modelling. Restricting our attention to the prospective studies of vertical transmission within each consortium, there are differences between the protocols that are described on page 8, lines 18-28

Line 47: Consider re-ordering the consortia so that they are listed in a consistent order when mentioned together throughout the document.

Good idea: we have revised to the order Alliance, PLAN, Action throughout

Line 54: "a higher proportion of adverse outcomes" (these are proportions, not rates; also, consider using language similar to this: "with higher incidence of Zika-associated birth defects" or "higher risk of Zika-associated birth defects in the first trimester.")

The revised manuscript uses this language

Page 6

Line 31: Should be noted (or expressed as a limitation) that IgM is not confirmatory.

The point being made here is that the vertical transmission rates being reported are all based on different tests and testing schedules. In the context of congenital infection – which is the focus of this section, our inclination at this time (see Supplementary Table S2) is to regard this as strongly indicative of congenital infection. This accords with references 8, 9, 14. However, this is provisional. The criteria for congenital infection will be determined by the Joint Diagnostics Group.

Line 36: Which tests? Serologic tests generally have limited specificity (particularly in the context of flavivirus cross-reactivity. But high specificity/sensitivity are characteristics of NAATs.

We are referring here to IgM and NAAT as markers of *congenital* infection. In this context cross-reactivity is less of an issue if – as is widely accepted – the other flaviviruses are not vertically transmitted across the placenta. While NAATs are analytically sensitive, , their *diagnostic sensitivity* for congenital infection is limited in neonatal samples collected at birth, as evidenced by the high proportion of CZS cases with no NAAT evidence of acute ZIKV congenital infection (references 13,14). Clearance of fetal infection prior to delivery has been directly observed (references 15,16). The diagnostic difficulties are fully described page 8 line 51 to page 9 line 40. This introductory section aims only to give some background to the literature as reported, and point out some of the difficulties involved in interpreting it.

Page 7

Line 39 (Objective 2): Specify what is meant by deliveries to women? Not all products of conception will be liveborn, so it might be better to specify "among fetuses and liveborn infants" instead of "deliveries" (or something along those lines)

Thank you: we have adopted this suggestion

Page 8 Line 3: Please specify how infection status during pregnancy will be ascertained. (It may differ between studies – if so, please mention this.)

This will be one of the responsibilities of the Joint Diagnostics Group described on page 9. This group will apply the same criteria to all the consortia, as far as that can be operationalized given the different data. It is difficult to be more specific at this stage, particularly as many of the samples are still being looked at with assays that did not exist when the study was designed. Provisional definitions, which are necessarily somewhat generic, are proposed in Supplementary Table S2.

Page 9 Line 20: Can you specify laboratory markers of vertical infection that will generally be used? As noted, a final determination of definitions based on available data will be elaborated by the Joint Diagnostic Group but, generally speaking, PCR or IgM evidence will be used.

Line 51: Generally, serologic tests (after ruling out DENV cross-reactivity) would indicate probable infection, although timing of infection cannot be inferred (and, as you aptly state, would not necessarily indicate an infection that occurred during pregnancy). Because of implications of “confirmed” infection (implying NAAT), I would suggest re-wording. Something along the lines of, “These tests therefore indicate probable (or presumptive) Zika or flavivirus infection, although timing of infection with regard to pregnancy cannot be ascertained.”

We cannot see exactly where you suggest this sentence should go, as Line 51 does not relate specifically to serologic tests. However, lines 28-30, we make exactly this point: “*These tests therefore indicate suspected but not confirmed MIP.*”

Page 10

Line 6: Suggest removing the sentence, “An IgG negative response in the woman shortly after delivery can be taken as suggestive of No MIP.” This does not rule out infections later in pregnancy, from which an IgG response may not have yet mounted.

We accept that an IgG negative test at delivery would not entirely rule out an infection very late in pregnancy. On the other hand we have data on some women a few weeks after delivery, which would allow a more definite conclusion. We think the words “suggestive of” express an appropriate degree of caution.

Pages 10

Lines 12-13: Suggest maintaining adherence to laboratory classifications established by WHO or CDC case definitions, particularly given the cross-cutting nature of these consortia. Confirmed is generally reserved for NAAT or IgM with positive ZIKV PRNT; probable is reserved for ZIKV IgM assays with either a negative DENV IgM or where no PRNT was performed. Suspected indicates clinical criteria and epidemiologic linkage criteria fulfilled. Please consider following this classification scheme throughout.

Thank you for this suggestion. We are sure that our expert Diagnostics Group will wish to adhere to established CDC or WHO definitions wherever possible, while recognising that requirements of individual diagnosis are very different from epidemiological studies. At this stage we are reluctant to commit the Group to any specific set of definitions

Page 11

Line 41: (General comment) Make sure that means of defining microcephaly are standardized across consortia, if possible.

Agreed, this is in hand.

Page 15

(For Discussion section): It’s important to note that the timeframe for data collection for two of the consortia (ZIKAction and ZIKAlliance) began after the peak of the Zika epidemic, as included studies were initiated in mid-2017 while peak transmission in the Americas occurred in mid-2016. This may affect sample size relative to ascertainment of laboratory confirmation of cases and subsequent outcomes.

Agreed: the sample size is indeed less than we had hoped for, due to the relatively late start of the projects on the ground. However, we would be reluctant to imply that this will detract from the value of our planned analysis because the joint analysis is intended to maximize the use of all available information.

General comment for tables: Consider revising tables to reflect standard laboratory-based case definitions (confirmed, probable, and suspected cases), as described previously.

General comment for supplementary materials: Information on serology testing is a little vague. Could you specify any anticipated follow-up testing to rule out serologic cross-reactivity with DENV (e.g., IgM ELISA or PRNT).

Unfortunately it is not possible to be more specific about how cross-reactivity will be ruled out.

Consortia have used different tests, and confirmatory and follow-up testing, including PRNT, is still underway. The Joint Diagnostics Group will have responsibility for devising criteria for the classification, and we are reluctant to prejudge these issues in this analysis protocol. On the labels themselves, although “confirmed, probable, suspected” can be considered as standard for laboratory

based definitions, papers on ZIKV have used a wider range of terms, and the epidemiological context – such as the incidence of DENV – could also be taken into account.